# Insights into the Multi-Azole Resistance Profile in *Candida haemulonii* Species Complex

**DOI:** 10.3390/jof6040215

**Published:** 2020-10-11

**Authors:** Laura Nunes Silva, Lívia de Souza Ramos, Simone Santiago Carvalho Oliveira, Lucas Barros Magalhães, Eamim Daidrê Squizani, Lívia Kmetzsch, Marilene Henning Vainstein, Marta Helena Branquinha, André Luis Souza dos Santos

**Affiliations:** 1Laboratório de Estudos Avançados de Microrganismos Emergentes e Resistentes (LEAMER), Departamento de Microbiologia Geral, Instituto de Microbiologia Paulo de Góes (IMPG), Universidade Federal do Rio de Janeiro (UFRJ), Rio de Janeiro 21941-901, Brazil; lauransilva@gmail.com (L.N.S.); liviaramos2@yahoo.com.br (L.d.S.R.); simonesantiagorj@yahoo.com.br (S.S.C.O.); lbarrosmagalhaes@gmail.com (L.B.M.); mbranquinha@micro.ufrj.br (M.H.B.); 2Centro de Biotecnologia, Universidade Federal do Rio Grande do Sul (UFRGS), Porto Alegre, Rio Grande do Sul 91540-000, Brazil; eamimsquizani@gmail.com (E.D.S.); liviak@cbiot.ufrgs.br (L.K.); mhv@cbiot.ufrgs.br (M.H.V.); 3Programa de Pós-Graduação em Bioquímica (PPGBq), Instituto de Química (IQ), Universidade Federal do Rio de Janeiro (UFRJ), Rio de Janeiro 21941-909, Brazil

**Keywords:** azole resistance, efflux pumps, fluconazole, lanosterol 14α-demethylase, non-*albicans Candida* species, voriconazole

## Abstract

The *Candida haemulonii* complex (*C. duobushaemulonii*, *C. haemulonii*, and *C. haemulonii* var. *vulnera*) is composed of emerging, opportunistic human fungal pathogens able to cause invasive infections with high rates of clinical treatment failure. This fungal complex typically demonstrates resistance to first-line antifungals, including fluconazole. In the present work, we have investigated the azole resistance mechanisms expressed in Brazilian clinical isolates forming the *C. haemulonii* complex. Initially, 12 isolates were subjected to an antifungal susceptibility test, and azole cross-resistance was detected in almost all isolates (91.7%). In order to understand the azole resistance mechanistic basis, the efflux pump activity was assessed by rhodamine-6G. The *C. haemulonii* complex exhibited a significantly higher rhodamine-6G efflux than the other non-*albicans Candida* species tested (*C. tropicalis*, *C. krusei*, and *C. lusitaneae*). Notably, the efflux pump inhibitors (Phe-Arg and FK506) reversed the fluconazole and voricolazole resistance phenotypes in the *C. haemulonii* species complex. Expression analysis indicated that the efflux pump (*ChCDR1*, *ChCDR2*, and *ChMDR1*) and *ERG11* genes were not modulated by either fluconazole or voriconazole treatments. Further, *ERG11* gene sequencing revealed several mutations, some of which culminated in amino acid polymorphisms, as previously reported in azole-resistant *Candida* spp. Collectively, these data point out the relevance of drug efflux pumps in mediating azole resistance in the *C. haemulonii* complex, and mutations in ERG11p may contribute to this resistance profile.

## 1. Introduction

The recently described *Candida haemulonii* clade has caused deep concerns in hospital environments worldwide, since its members are able to cause life-threatening infections presenting high rates of clinical treatment failures [1]. The *C. haemulonii* clade is composed of emerging, opportunistic, and multidrug-resistant species formed by *C. auris*, *C. duobushaemulonii*, *C. haemulonii sensu stricto*, *C. haemulonii* var. *vulnera*, *C. pseudohaemulonii*, and *C. vulturna* [1,2].

*Candida duobushaemulonii*, *C. haemulonii*, and *C. haemulonii* var. *vulnera*, which form the *C. haemulonii* complex, have been identified in different geographic regions in the globe, prominently in Latin American (e.g., Brazil) and Asian (e.g., India and China) countries. However, not surprisingly Coles et al. [3] highlighted the possible emergence of these fungal pathogens also in the United States of America. This fungal complex is often associated with bloodstream infections, onychomycosis, vaginal infections, catheter-related fungemia, and outbreaks in neonatal intensive care units [3,4,5,6,7,8,9]. In Brazil, most of the studies reported candidemia by the *C. haemulonii* species complex, associated with high antifungal resistance profiles [10,11,12,13]. Since phenotypic identification methods cannot accurately distinguish among these species, the prevalence of *C. haemulonii* species complex in hospital settings may be underestimated. In this sense, a 10.5-year analysis conducted between 1997 and 2007 by the ARTEMIS DISK Global Antifungal Surveillance Study demonstrated that infections caused by *C. haemulonii* were rare (less than 0.01%) at that moment [14]. In 2012, *C. haemulonii* was identified as the third cause of candidemia in a hospital in New Delhi, India, corresponding to 15.5% of the cases [15]. Lately, Xiao et al. [16] reported surveillance results on candidemia from 77 Chinese hospitals, where the prevalence of *C. haemulonii* rose from 0.6% to 0.9% in the 3-year investigation period (2015–2017). Additionally, a recent study from Brazil demonstrated an increased prevalence of the *C. haemulonii* species complex from 0.9% in the first period of the analysis (2008-2013) to 1.7% in the second period (2014-2019) [17].

The *C. haemulonii* species complex typically demonstrates resistance to amphotericin B (AMB) and fluconazole as well as a variable susceptibility to the novel azoles (e.g., voriconazole, posaconazole, and isavuconazole) and echinocandins (e.g., caspofungin, micafungin, and anidulafungin) [18,19,20]. Recently, our research group described primary evidence suggesting that resistance to AMB is due to multifactorial events displayed by the species belonging to the *C. haemulonii* complex, including reduced ergosterol content located in the plasma membrane, mitochondrial dysfunction, resistance to oxidative stressors, and the upregulation of the antioxidant machinery [21].

Fluconazole is the most widely used antifungal agent in hospital settings for both treatment and prophylaxis approaches. However, fluconazole resistance is particularly prominent in the *C. haemulonii* species complex, with most of the clinical isolates presenting a typical resistant profile to this triazole antifungal [18,20]. The main mechanism of action of azole antifungals involves the blockage of the ergosterol synthesis pathway in fungal cells, specifically by inhibiting the lanosterol 14α-demethylase, an enzyme encoded by the *ERG11* gene. The lanosterol 14α-demethylase (ERG11p) inhibition leads to the accumulation of aberrant sterol intermediates at the plasma membrane, inducing severe toxic effects that culminate in the interruption of fungal growth. Mutations in the *ERG11* gene are one of the most important mechanisms described for azole resistance in *Candida* spp. [22]. Nevertheless, it is well known that a fungus can orchestrate more than one mechanism of action in order to become resistant to azoles, including the upregulation of the *ERG11* gene and the overexpression of efflux pumps mainly encoded by multidrug-resistance (MDR) and candida drug resistance (*CDR*) genes [22]. Although recent studies have suggested that mutations in the *ERG11* gene lead to the production of altered ERG11p in *C. auris* [23,24], an analysis of the genomic and transcriptomic data of *C. haemulonii* clade also impute the upregulation of the *CDR1* gene as an important mechanism explaining azole resistance [23,24,25].

Thus far, no *C. auris*, *C. pseudohaemulonii*, and *C. vulturna* isolates has been described in Brazil. Since many reports have suggested intrinsic resistance to fluconazole and in vitro multidrug-resistance to azoles, we have examined the above-mentioned resistance mechanisms among 12 Brazilian clinical isolates of the *C. haemulonii* species complex. To accomplish this, we have evaluated the (i) in vitro susceptibility to different azoles; (ii) the efflux pump activity; (iii) the expression level of *ERG11*, *MDR*-like, and *CDR*-like genes; and (iv) mutations in the *ERG11* gene, which support the alterations in ERG11p that is the azoles’ target.

## 2. Materials and Methods

### 2.1. Fungal Isolates and Culture Conditions

Twelve clinical isolates of the *C. haemulonii* complex were used in this study: *C. haemulonii* (LIP*Ch*2, LIP*Ch*3, LIP*Ch*4, LIP*Ch*7, and LIP*Ch*12), *C. duobushaemulonii* (LIP*Ch*1, LIP*Ch*6, LIP*Ch*8, and LIP*Ch*10), and *C. haemulonii* var. *vulnera* (LIP*Ch*5, LIP*Ch*9, and LIP*Ch*11), identified by *ITS* gene sequencing, as previously described by our research group [10]. For a comparative purpose, three reference non-*albicans Candida* species were also used: *C. tropicalis* (ATCC 750), *C. krusei* (ATCC 6258), and *C. lusitaniae* (ATCC 200950). In all experiments, Sabouraud-dextrose medium was used to culture the fungal isolates at 37 °C for 48 h under constant agitation (200 rpm). Yeast cells were counted using a Neubauer chamber.

### 2.2. Antifungal Susceptibility Assay

Antifungal susceptibility testing was performed according to the standardized broth microdilution technique described by the Clinical and Laboratory Standards Institute (CLSI) in document M27-A3 and interpreted according to document M27-A3 [26]. The tested antifungal drugs include fluconazole (FLC), itraconazole (ITC), ketoconazole (KTC), posaconazole (PSC), and voriconazole (VRC) (Sigma-Aldrich, USA). The minimum inhibitory concentration (MIC), modal MIC, geometric mean (GM)-MIC, MIC_50_, MIC_90_, median, and antifungal concentration range were calculated using Prism version 8 (GraphPad Software, USA). Until now, no species-specific clinical breakpoints are established for rare species regarding azole susceptibility both in CLSI and EUCAST. In this sense, the manuscripts focused on this fungal complex usually base their results on the CLSI breakpoints established for the *Candida* genus (CLSI document M27-A3) in order to have a minimum (even not precisely) parameter to interpret these data. The clinical breakpoints defined for *Candida* spp. were used for the interpretation of the MIC data as follows: susceptible (S) ≤8 mg/L, susceptible-dose dependent (S-DD) 16–32 mg/L, resistant (R) ≥64 mg/L for FLC; S ≤0.125 mg/L, S-DD 0.25–0.5 mg/L, R ≥1 mg/L for ITC; S ≤1 mg/L, S-DD 2 mg/L, R ≥4 mg/L for VRC. For PSC, the susceptibility breakpoint was considered 0.06 mg/L, determined by EUCAST for *Candida* spp. No interpretive criteria for KTC are available in either the CLSI or EUCAST documents.

### 2.3. Efflux Pump Activity

To assess the ABC-type drug transporter activity of fungal cells, we evaluated the glucose-induced efflux of rhodamine 6G (R6G) (Sigma-Aldrich, USA) assay, as previously described [6,27]. Yeasts (10^7^ cells/mL) were incubated with R6G (10 µM) to enable uptake under carbon source-depleted conditions for 60 min. Cells were washed in phosphate-buffered saline (PBS), and tubes with glucose-free PBS (control) and PBS supplemented with 100 mM of glucose were prepared to start R6G efflux. Finally, after 1 and 3 h of incubation with glucose at 37 °C, the cells were quantified in a flow cytometer (FACSCalibur, BD Bioscience, USA) and analyzed at 527 nm. Representative data from the 10,000 events analysis were shown and the results were expressed as the percentage of fluorescent cells (%FC). Fungal cells not stained with R6G were used as a negative control. In parallel, a different group was also pretreated for 1.5 h with FLC (64 mg/L) and VRC (16 mg/L) before the addition of R6G and glucose.

### 2.4. Chemosensitization Assay Combining Efflux Pump Inhibitors and Azoles

The efflux pump inhibitors (EPIs), Phe-L-Arg-β-naphthylamine dihydrochloride (Phe-Arg) and tacrolimus (FK506) (Sigma-Aldrich, USA), were used in combination with FLC or VRC in order to determine if the antifungal activity could be enhanced [28]. MICs were developed in the presence of azole (64 to 0.125 mg/L) with and without the presence of each EPI at a concentration of 64 mg/L. All the systems were incubated for 48 h at 37 °C and the results were expressed as the lowest azole concentration that resulted in the total inhibition of visible fungal growth.

### 2.5. RNA Extraction

After initial culture conditions (item 2.1), *C. haemulonii* isolate LIP*Ch*4 (a pan-azole *C. haemulonii sensu strictu* isolate) was diluted to an initial inoculum of 10^7^ cells/mL in fresh Sabouraud (20 mL) and then exposed to FLC at 64 mg/L or VRC at 16 mg/L for 1.5 h with shaking. Following drug exposure, the fungal cells were harvested for RNA isolation as previously described [29]. Complementary DNA (cDNA) was synthesized with the High-Capacity cDNA Reverse Transcription Kit (Applied Biosystems™, Thermo Scientific, USA).

### 2.6. Quantitative Reverse Transcription-Polymerase Chain Reaction (RT-qPCR)

RT-qPCR was undertaken to estimate the expression of the *ChCDR1*, *ChCDR2*, *ChMDR1*, and *ChERG11* ortholog genes by *C. haemulonii* complex strains during FLC or VRC exposure (after 1.5 h of exposure). cDNA was analyzed by RT-qPCR with the StepOne™ Real-Time PCR System (Applied Biosystems™, Thermo Scientific, USA) and specific primers (Table 1). Universal SYBR green Supermix was used for PCRs according to the manufacturer’s recommendations. The 2^−ΔΔCT^ method was used for the relative quantification of gene expression [30], and the data were normalized to the *ACT1* (actin) gene expression.

### 2.7. Amplification and Sequencing of the ERG11 Gene

The *ERG11* gene encoding lanosterol 14α-demethylase was amplified and sequenced with specific primers (Appendix A). The BigDye^®^ Terminator v3.1 Cycle Sequencing Kit (Applied Biosystems, USA) was used for sequencing the reaction, precipitated with ethanol/EDTA/sodium acetate according to the protocol suggested by the manufacturer and sequenced on the ABI3730xl DNA Analyzer platform (Applied Biosystems, USA). DNA sequences and the corresponding amino acid sequences were analyzed with the SeqMan II and Bioedit software packages (Lasergene; DNAStar, USA). Consensus sequences were aligned with different reference *ERG11* sequences from *Candida* spp. For the mutation analysis, the *C. albicans* strain SC5314 *ERG11* gene assembly sequence from Candida Genome Database (http://www.candidagenome.org/) was used [31,32,33].

### 2.8. Sequences Accessions

The *ERG11* gene sequences obtained from clinical isolates belonging to the *C. haemulonii* species complex studied herein were deposited in GenBank with the following accession numbers: MT860384 to MT860395.

### 2.9. Statistics

All the experiments were performed in triplicate, in three independent experimental sets. The results were analyzed statistically by the Analysis of Variance One-Way ANOVA (comparisons between three or more groups). All the analyses were performed using the program GraphPad Prism8. In all analyses, *p* values of 0.05 or less were considered statistically significant.

## 3. Results

### 3.1. Azoles’ Susceptibility Profiles

The MIC values determined for 12 clinical isolates forming the *C. haemulonii* species complex (*C. haemulonii*, *n* = 5 isolates; *C. duobushaemulonii*, *n* = 4; and *C. haemulonii* var. *vulnera*, *n* = 3) against five different azoles (FLC, ITC, KTC, PSC, and VRC) were summarized in Table 2 and Appendix A. Among the tested azoles, PSC (GM-MIC = 0.62 mg/L) and KTC (GM-MIC = 1.49 mg/L) exhibited potent antifungal activity against all the isolates studied. Notably, all the fungal isolates had MICs of ≥64 mg/L for FLC. For ITC, a modal MIC of ≥16 mg/L was noted, suggesting that all the isolates were resistant to this antifungal. Despite some isolates remaining susceptible to VRC, the MIC distribution exhibited a peak at 16 mg/L, with 66% (*n* = 8) of isolates being resistant to this azole. Nine isolates (75%) were resistant to at least two different azoles. Six isolates (50%) were considered pan-azole resistant, of which four belong to the species *C. haemulonii senso strictu*, one belongs to *C. duobushaemulonii*, and one belongs to *C. haemulonii* var. *vulnera*.

### 3.2. Efflux Pumps Play a Primary Role in Azole Resistance in C. haemulonii Complex

In this set of experiments, we firstly assessed the efflux pump activity by the extrusion of RG6. When compared with other non-*albicans Candida* species (*C. tropicalis, C. lusitaniae*, and *C. krusei*), *C. haemulonii* exhibited a significantly higher R6G efflux after 1 h (Figure 1) (Appendix A). Analyzing the glucose-induced R6G efflux in the three different species forming the *C. haemulonii* complex, we could not observe statistical differences between them over 3 h (Figure 2). Interestingly, the efflux pump activities have been shown to be constitutively expressed within fungal cells, since the treatment with either FLC or VRC did not modulate the glucose-induced efflux of RG6 (Figure 2).

Since it seems that efflux pump activities are constitutively present in the *C. haemulonii* complex, as the FLC and VRC treatment did not modify the R6G extrusion process, we next examined the expression of the genes coding for known azole transporters. In an effort to identify the efflux pump-encoding genes which could participate in the clinical azole resistance in the *C. haemulonii* species complex, *CDR1*-like, *CDR2*-like, and *MDR1*-like genes were identified by the BLAST approach using the recently updated genome of *C. haemulonii* (strain B11899), which presented the highest degree of homology genes to *C. albicans*. The *C. haemulonii* genes with the highest degree of homology to the *C. albicans CDR1* and *CDR2* genes were XM_025483931.1 and XM_025488660.1, here referred to as *ChCDR1* and *ChCDR2* (Appendix A), respectively; the *C. haemulonii* gene with a greater degree of homology with the *C. albicans MDR1* gene was XM_025488295.1, here referred to as *ChMDR1* (Appendix A). Our data showed that the *ChCDR1, ChCDR2*, and *ChMDR1* genes were not modulated by the azole treatment (Figure 3). A similar result was observed considering the expression of the *ERG11* gene (Figure 3).

We then assessed the contribution of these transporters to azole sensitivity by blocking them with classical EPIs (Table 3). When the isolates were grown in the presence of a combination of an azole (FLC or VRC) plus an EPI (Phe-Arg or FK506), we can observe that the MIC decreased in a 4- to more than 64-fold range. Remarkably, all the tested fungal isolates reversed their resistance phenotypes with the addition of EPIs.

### 3.3. ERG11 Mutations and Gene Expression Analysis

A phylogenetic tree of *ERG11* genes and their putative orthologs in *Candida* spp. was built by the MEGAX program using the “Maximum likelihood” method, with the intent to identify their phylogenetic relationships (Figure 4). The tree based on a concatenated alignment of *ERG11* genes places *C. auris, C. haemulonii, C. duobushaemulonii*, and *C. pseudohaemulonii* as a single clade, confirming the close relationship among these fungal species. Our phylogenetic analysis strongly supported that *C. duobushaemulonii* and *C. pseudohaemulonii* were more closely related and form a sister group of *C. haemulonii*, which appeared as the most branched species [23].

By comparing the amino acid sequences of ERG11p, 12 amino acid substitutions were identified in our isolates, all of them have been previously listed in the hot spot (HS) regions I, II, and III of ERG11p of *C. albicans* and *C. auris* (Figure 5) [23,35,36]. Of the 12 found substitutions in ERG11p, modifications at positions K119S were only present in *C. haemulonii sensu stricto* and *C. haemulonii* var*. vulnera*, whereas changes at positions R267T and A432S were only detected in *C. duobushaemulonii* isolates.

## 4. Discussion

Drug resistance has become a major public health issue in terms of managing mainly invasive infections, particularly those resulting from pathogenic fungi that present scarce available effective drugs for treatment. Despite still being considered rare, infections by the *C. haemulonii* species complex have been highlighted by the high capability of developing multidrug-resistance against all clinically available antifungal classes, representing a challenge to the treatment of patients affected by these fungi. The majority of the studies has shown that isolates comprising the *C. haemulonii* complex demonstrated high MIC values for AMB and also frequently to azoles, all directly associated with clinical treatment failure [18,20]. In general, susceptibility to echinocandins, such as caspofungin and micafungin, in both in vitro and in vivo approaches have been observed [6,37,38]. However, there are already reports of resistance to this “novel” antifungal class in clinical isolates belonging to the *C. haemulonii* species complex [11,12]. Notably, we remain at the very beginning in understanding the biology of these “rare” fungal opportunistic human pathogens.

Among the diversity of molecular mechanisms related to the antifungal resistance machinery orchestrated by fungal cells, the upregulation of the efflux pumps *MDR1* and/or *CDR1/CDR2* contributes to azole resistance in most clinical isolates of *Candida* spp. [39]. Our results align with Zhang et al. [24], who observed that the presence or absence of FLC treatment repeatedly demonstrated the upregulation of the multidrug transporter gene *CDR1*. Additionally, in that study, treatment of the azole-resistant *C. haemulonii* strains with EPIs partially restored their susceptibility to FLC, confirming the significant contribution of Cdr1p to azole resistance [24]. It is well known that EPIs, such as Phe-Arg and FK506, play a vital role in azole resistance reversal against *C. albicans* [40], *C. glabrata* [41], *C. tropicalis* [42], and *C. auris* [28]. Consistent with these findings, in the present study, the combination of Phe-Arg or FK506 with FLC or VRC showed synergism against all the resistant isolates forming the *C. haemulonii* complex. Hence, here we have provided important evidence of efflux pumps ruling the azole resistance phenotype in the *C. haemulonii* species complex.

For many years, the imidazole KTC was the only available oral agent for the treatment of systemic fungal infections. Nevertheless, the side effects associated with KTC therapy as well as the the relatively poor response rates, led to the search for a novel chemical group of azole derivatives, namely the triazoles [43]. Overall, the triazoles demonstrate a broader spectrum of antifungal activity and reduced toxicity when compared with the imidazole derivatives [44]. PSC is a second-generation triazole agent that has been developed mainly to combat the appearance of azoles resistance in yeasts, in particular to FLC. Several in vitro and in vivo studies confirmed that PSC has a broad spectrum of activity against the majority of yeasts, filamentous fungi and azole-resistant *Candida*. Thus, PSC has a superior activity profile when compared with FLC and ITC [44]. For instance, one way of developing azole resistance is by acquiring mutations on the *ERG11* gene. Depending on the position and the nature of the alteration, the reduced susceptibility can cause cross-resistance or just resistance to a subset of derivatives [44].

Although the role of *ERG11* gene point mutations in azole resistance is a well-recognized phenomenon in *Candida* spp. [38], only recently a few studies have characterized the polymorphisms of the *ERG11* gene in azole-resistant isolates of the *C. hameulonii* complex [2,13]. In the study of Gade et al. [2], an increasing azole resistance in isolates of *C. haemulonii* and *C. duobushaemulonii* was evidenced in Colombia, Panama, and Venezuela, associated with substitutions in the *ERG11* gene, suggesting the existence of an evolutionary pressure for the development of azoles resistance in Latin America. In the present study, eight of the 12 *ERG11* gene mutations have been reported to be associated with reduced azole susceptibility in *C. albicans*—namely, F105L, S110A, D116A, K119S, D153E, R267T, A432S, and F487Y. However, Erg11p substitutions F126L, Y132F, and K143R, which were implicated in reduced susceptibilities to azoles upon heterologous expression in *S. cerevisiae* [45,46], were not identified in our isolates. Corroborating our data, a recent study by Rodrigues et al. [13] again found that Brazilian resistant isolates of *C. haemulonii* var*. vulnera* also lacked these common mutations. Thus, the *ERG11* gene mutations by themselves cannot explain the differences between the distinct multi-azole profile in our isolates. *ERG11*, *ERG3*, and *ERG5* combinatorial alterations in *Candida* have shown to circumvent this pathway and led to the development of cross-resistance to azoles and AMB [47,48,49,50]. Using multi-omics approaches, Zamith-Miranda et al. [51] showed that a resistant isolate of *C. auris* displays a significant higher abundance of Erg2p and a lower abundance of Erg3p. As a matter of fact, we have demonstrated by GC-MS that the *C. haemulonii* species sterol profiles were comparable to those of different *Candida* species with *ERG2*, *ERG3*, *ERG6*m and *ERG11* gene mutations, presenting the same cross-resistance profile [21]. Along with our results, recent literature reports have suggested that azole resistance in *C. haemulonii* clade can be related to multiple-steps modifications in the ergosterol biosynthesis pathway [45,51].

Azole resistance has also been attributed to fungal cells that have lost mitochondrial function resulting in respiratory deficiency [52,53,54,55]. Precisely, in *S. cerevisiae*, *C. glabrata*, and *A. fumigatus*, mitochondrial dysfunction has displayed multiple effects in fungal cells, and this can enlighten us on the enhanced efflux pumps’ activities [53,56,57]. These fungal cells usually exhibit an altered sterol profile with a unequal amount of ergosterol; still, no changes in the sequence of *ERG11* gene or its expression have been detected [57,58]. Our recently published work demonstrated primary evidence that impariment of the mitochondrial function in *C. haemulonii* species complex intensifies the tolerance to stress and could be related with a rearrangement in the cellular ergosterol content [21]. In this context, growing evidence from recent elegant reports has revealed that there are more complicated and unknown drug resistance mechanisms apart from the mutational modification of the azole target ERG11p in fungal pathogens [59,60,61,62].

## 5. Conclusions

Indeed, the widespread use of azoles has been considered to be an important risk factor for FLC resistance development primarily in non-*albicans Candida* species. In this context, an assessment of all possible resistance mechanisms is important for a proper patient treatment outcome. Our data suggest an emergence of azole-resistant clinical isolates belonging to the *C. haemulonii* species complex recovered from Brazilian patients. This study provides significant evidence that efflux pump activity is implicated in high-level FLC resistance and pan-azole resistance in the *C. haemulonii* complex. Once there is an increasing azole resistance among *C. haemulonii* complex isolates, further surveillance is warranted.

## Figures and Tables

**Figure 1 jof-06-00215-f001:**
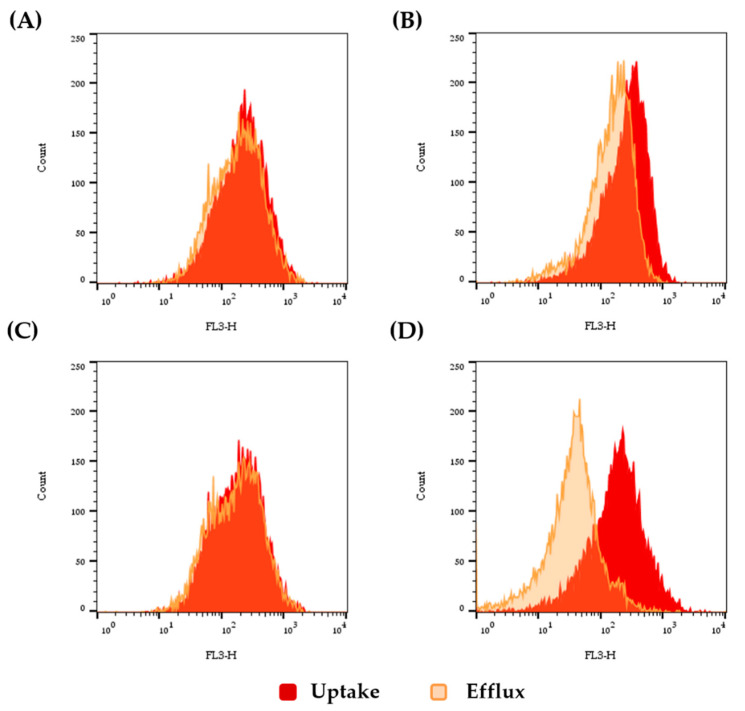
Flow cytometry analysis of the glucose-induced rhodamine 6G (R6G) efflux. Uptake of fluorochrome was quantified by incubating yeast cells in PBS for 60 min with 10 µM of R6G (red curve). Next, efflux was evaluated by quantifying the residual fluorescence of the cells after the removal of the free dye and with an additional incubation in phosphate-buffered saline (PBS) supplemented with 100 mM of glucose (orange curve). Fungal species: (**A**) *C. tropicalis* (ATCC 750), (**B**) *C. krusei* (ATCC 6258), (**C**) *C. lusitaniae* (ATCC 200950), and (**D**) *C. haemulonii* (LIPCh4). Each experiment was performed at least for three independent times with similar results.

**Figure 2 jof-06-00215-f002:**
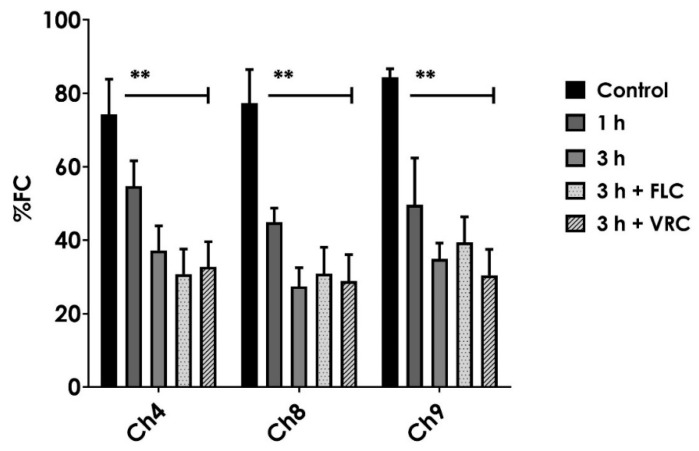
Flow cytometric efflux of rhodamine 6G (R6G) over time among clinical isolates belonging to the *C. haemulonii* species complex. Data were expressed as the percentage of fluorescent cells (%FC) after the addition of 100 mM of glucose. Some systems were also pretreated for 1.5 h with fluconazole (FLC at 64 mg/L) and voriconazole (VRC at 16 mg/L) prior to the R6G and glucose supplementation. Fungal species: *C. haemulonii LIPCh4* (Ch4), *C. duobushaemulonii* LIP*Ch*8 (Ch8), and *C. haemulonii* var. *vulnera* LIP*Ch*9 (Ch9). The results were expressed as the mean ± standard deviation of three independent experiments, and the symbol (**) represents *p* values ≤ 0.01 when compared to control (one-way ANOVA, Bonferroni post hoc test).

**Figure 3 jof-06-00215-f003:**
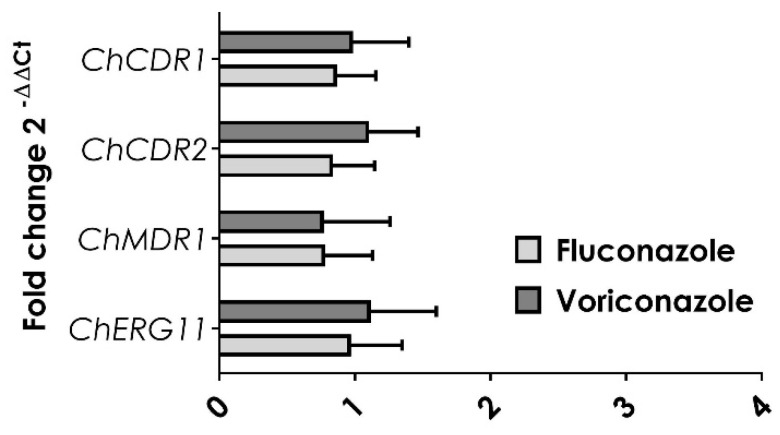
Expression of the *CDR1*, *CDR2, MDR1*, and *ERG11* genes in *C. haemulonii*. The expression of target genes was determined by RT-qPCR in the clinical isolate of *C. haemulonii* LIP*Ch*4. A pre-treatment was conducted 1.5 h earlier with fluconazole (FLC at 64 mg/L) and voriconazole (VRC at 16 mg/L) before the RNA extraction. Actin gene (*ACT1*) was used as a housekeeping gene. The results were expressed as the mean ± standard deviation of three independent experiments.

**Figure 4 jof-06-00215-f004:**
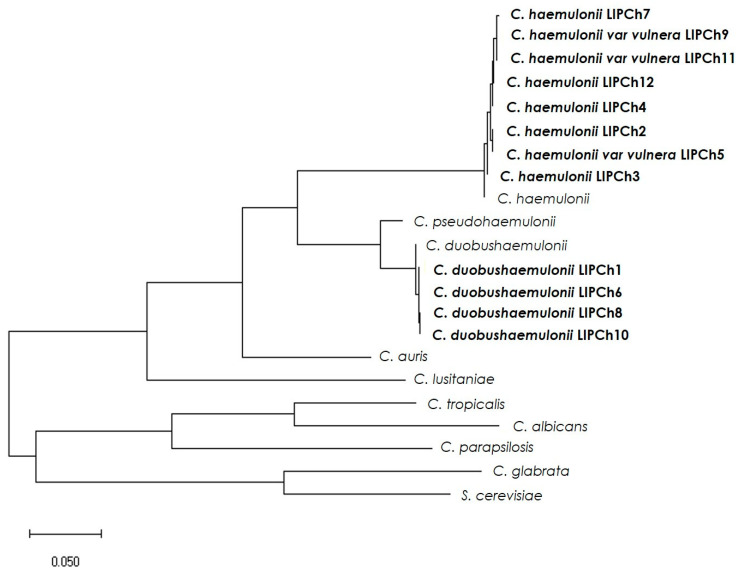
Phylogenetic tree of the *ERG11* gene from different *Candida* spp. The alignment was generated by the CLUSTAL W program and the phylogram was constructed using the “Maximum likelihood” method using the MEGAX software [34]. *ERG11* and its orthologous amino acid sequences (followed by NCBI access numbers in parentheses): *Saccharomyces cerevisiae* S288C lanosterol 14α-demethylase (NM_001179137.1), *Candida glabrata* ATCC 90030 lanosterol 14α-demethylase (KR998002.1), *Candida parapsilosis* ATCC 22019 lanosterol 14α-demethylase (GQ302972.1), *Candida albicans* SC5314 lanosterol 14α-demethylase (XM_711668.2), *Candida tropicalis* ATCC 750 lanosterol 14α-demethylase (KR998015.1), *Candida lusitaniae* CBS 6936 lanosterol 14α-demethylase (EU919443.1), *Candida auris* B11221 lanosterol 14α-demethylase (XM_029033208.1), *Candida duobushaemulonii* B09383 hypothetical protein (XM_025482007.1), *Candida pseudohaemulonii* B12108 hypothetical protein (XM_024860251.1), and *Candida haemulonni* 14α-demethylase (XM_025486744.1). The lengths of the branches indicate the average number of changes per site.

**Figure 5 jof-06-00215-f005:**
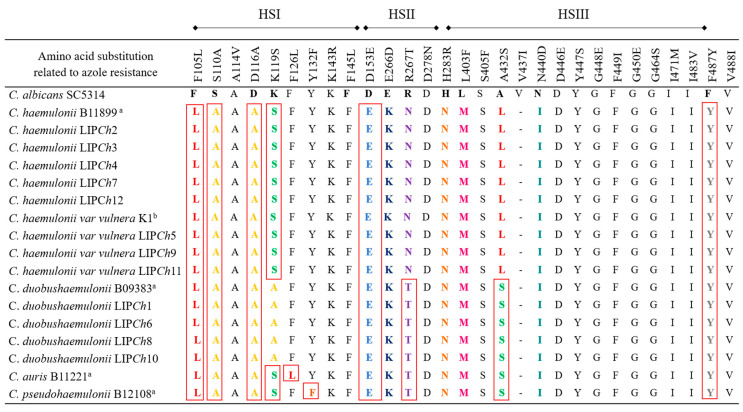
Multiple alignment of the azole drug target ERG11p. Observed amino acid substitutions in *C. albicans* compared with the clinical isolates of the *C. haemulonii* species complex. Red boxes indicate the observed azole resistance amino acids substitutions in the three different hotspots (HSI, HSII, and HSIII). Amino acid numbers are based on the *C. albicans* protein sequence. Previously published by ^a^ Muñoz et al. [23] and ^b^ Rodrigues et al. [13]. Colored letters are mutations encountered in the sequence alignment. Red boxes are mutation previously described in resistant strains of *C. albicans*.

**Table 1 jof-06-00215-t001:** Oligonucleotide sequences used in this study.

Oligonucleotides ^a^	Sequences (5′ to 3′)	Purpose—Accession Number ^b^
ERG11_CH_F1	CATTTTCAGGCCTTGCCCAC	*ERG11* sequencing *Candida haemulonii*—XM_025486744.1
ERG11_CH_R1	TCTGGGCACGTATCTCTGGA	
ERG11_CH_F2	ACTGCCTTAACCAAGGAGGC	
ERG11_CH_R2	AAGCTACCACCTTTGGAGGC	
ERG11_CH_F3	TTTTGGCCTCCAAAGGTGGT	
ERG11_CH_R3	CGCATGTCTCCCTCTTCTCC	
ERG11_CD_F1	CGGTATGCAGCCATACGAGT	*ERG11* sequencing *Candida duobushaemulonii* —XM_025482007.1
ERG11_CD_R1	CGCCAATCAACAAGTTGGCA	
ERG11_CD_F2	TCCAGAGATACGTGCCCAGA	
ERG11_CD_R2	ACACAGAGAGCACCTCGTTG	
ERG11_CD_F3	CTGCCTGGTTCTTGTTGCAC	
ERG11_CD_R3	AGTCAACAGGTGGAAGCGAG	
ACT1_CH_F1	ACTGCTTTGGCTCCATCCTC	*ACT1* real-time PCR—XM_025485061.1
ACT1_CH_R1	AGACTCGTCGTACTCCTGCT	
ERG11_CH_F1	CTGGATCCCATGGTTTGGCT	*ERG11* real-time PCR—XM_025486744.1
ERG11_CH_R1	GTCAAATGCGAGTAAGCGGC	
CDR1_CH_F1	ACTTGTCATGCCACGCAAAC	*CDR1* real-time PCR—XM_025483931.1
CDR1_CH_R1	GGTAGCGCCTCTCGTACTTC	
CDR2_CH_F1	ATCGAGACCGGTGAGAGTGA	*CDR2* real-time PCR—XM_025488660.1
CDR2_CH_R1	CACCAGACGCACCCATAAGT	
MDR1_CH_F1	GTCCCTTCGGTGCAAAAACC	*MDR1* real-time PCR—XM_025484012.1
MDR1_CH_R1	AGGGCTAGCAAAGAAGCCTG	

^a^ The letters F and R in the primer names describe the 5′-to-3′ orientations of the primers as follows: F, forward (sense); R, reverse (antisense). ^b^ NCBI reference sequence.

**Table 2 jof-06-00215-t002:** MIC distribution of the *C. haemulonii* species complex (*n* = 12 isolates) against five azole agents tested using the CLSI standard method.

Azoles	0.016	0.032	0.064	0.125	0.25	0.5	1	2	4	8	16	32	64	Range	GM-MIC ^a^	MIC_50_ ^b^	MIC_90_ ^c^
FLC													12	-	64	64	64
ITC					1	2	1			3	5			0.25–16	4.23	8	16
VRC					1	2	1				8			0.25–16	4.75	16	16
PSC		3		1	1	2	1			2	2			0.03–16	0.62	0.5	16
KTC				1		2	2	4	2		1			0.125–16	1.49	2	12.4

FLC, fluconazole; ITC, itraconazole; VRC, voriconazole; PSC, posaconazole; KTC, ketaconazole. MIC, minimum inhibitory concentration. ^a^ GM-MIC, geometric mean MIC. ^b^ MIC_50_, MIC at which 50% of the test isolates were inhibited. ^c^ MIC_90_, MIC at which 90% of the test isolates were inhibited. Modal MICs are indicated with underlined numbers.

**Table 3 jof-06-00215-t003:** Effect of the inhibition of efflux pumps (EPIs) on the azole susceptibility in clinical isolates of the *C. haemulonii* species complex.

Isolates	Azoles	MIC	MIC + Phe-Arg	Variation	MIC + FK506	Variation
LIP*Ch*4	FLC	>64	2	**≥32×**	8	**≥8×**
	VRC	>16	0.25	**≥64×**	1	**≥16×**
LIP*Ch*5	FLC	64	2	**≥32×**	4	**≥16×**
	VRC	0.5	0.0625	**8×**	0.125	**4×**
LIP*Ch*6	FLC	64	2	**32×**	4	**16×**
	VRC	1	0.0625	**8×**	0.25	**4×**
LIP*Ch*7	FLC	>64	2	**≥32×**	8	**≥8×**
	VRC	16	0.25	**64×**	1	**16×**
LIP*Ch*8	FLC	>64	4	**≥16×**	8	**4×**
	VRC	>16	0.5	**≥32×**	2	**8×**
LIP*Ch*9	FLC	64	4	**16×**	8	**8×**
	VRC	16	0.125	**64×**	0.5	**32×**

MIC, minimum inhibitory concentration; FLC, fluconazole; VRC, voriconazole. Phe-Arg and FK506, efflux pump inhibitors. Yeast species*: C. haemulonii* LIP*Ch*4, *C. haemulonii* var. *vulnera* LIP*Ch*5, *C. duobushaemulonii* LIP*Ch*6, *C. haemulonii* LIP*Ch*7, *C. duobushaemulonii* LIP*Ch*8, and *C. haemulonii* var. *vulnera* LIP*Ch*9.

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
