# Peer review of "Insights into the Multi-Azole Resistance Profile in Candida haemulonii Species Complex"

_jof, 2020, doi:10.3390/jof6040215_

Round 1

Reviewer 1 Report

Table 2.

The target of ketoconazole and posaconazole is Erg11, which is the same as ITCZ and FLCZ, but many strains tested were susceptible to ketoconazole and posaconazole. The reason should be discussed. Are ketoconazole and posaconazole less likely to be substrates for efflux pomp? Or are their Erg11 inhibitory effects stronger than itraconazole and fluconazole?

Ketoconazole is imidazole, not triazole.

Discussion

Is there a clinical report that the combined therapy of the efflux pump inhibitor, FK-506, and fluconazole has succeeded in infectious diseases caused by fluconazole-resistant Candida haemulonii?

How about clarifying that the triazole resistance of the C. haemulonii species complex used in this study is largely due to the increased expression of the efflux pumps and is less affected by the ERG11 mutation?

If there is a report that the ERG11 mutation contributed significantly to the high resistance of the C. haemulonii species complex, please describe it in the discussion.

Author Response

Reviewer 1: Table 2. The target of ketoconazole and posaconazole is Erg11, which is the same as ITCZ and FLCZ, but many strains tested were susceptible to ketoconazole and posaconazole. The reason should be discussed. Are ketoconazole and posaconazole less likely to be substrates for efflux pomp? Or are their Erg11 inhibitory effects stronger than itraconazole and fluconazole?

Authors: For many years, the imidazole KTC was the only available oral agent for the treatment of systemic fungal infections. Nevertheless, the side effects associated with KTC therapy as well as the relatively poor response rates, led to the search for a novel chemical group of azole derivatives, namely the triazoles. Overall, the triazoles demonstrate a broader spectrum of antifungal activity and reduced toxicity when compared with the imidazole antifungals [Maertens JA. History of the development of azole derivatives. Clinical Microbiology and Infection 2004; 10: 1-10.]. PSC is a second-generation triazole agent that has been developed mainly to combat the appearance of azoles resistance in yeasts, in particular to FLC. Several in vitro and in vivo studies confirmed that PSC has a broad spectrum of activity against the majority of yeasts, filamentous fungi and azole-resistant Candida species. Thus, PSC has a superior activity profile when compared with FLC and ITC [Zavrel M, Esquivel BD, White TC. The ins and outs of azole antifungal drug resistance: molecular mechanisms of transport. In: Berghuis A, Matlashewski G, Wainberg MA et al., eds. Handbook of Antimicrobial Resistance. New York, NY: Springer New York, 2017; 423-52].

For instance, one way of developing azole resistance is by acquiring mutations on ERG11 gene. Depending on the position and the nature of the alteration, the reduced susceptibility can cause cross-resistance or just resistance to a subset of derivatives2. Until now, we don’t have a clue about the differences in azole susceptible profile of our isolates since the mutations are presented in all isolates. Maybe in a next work, we can prove by molecular docking simulations a better explanation for this profile.

All these explanations are added to the revised manuscript version.

Reviewer 1: Ketoconazole is imidazole, not triazole.

Authors: We are grateful for the correction. We modified the sentences in the revised version of the manuscript. 

Reviewer 1: Discussion. Is there a clinical report that the combined therapy of the efflux pump inhibitor, FK-506, and fluconazole has succeeded in infectious diseases caused by fluconazole-resistant Candida haemulonii?

Authors: Candida haemulonii complex is an emergent group of MDR (Multidrug-Resistant) yeasts, that so far little is known. Clinicians tend to start an empirical treatment firstly with fluconazole or voriconazole and latest with caspofungin. Until now, no combination therapy was demonstrated. In this way, our research group is looking for this possibility and some experiments in this way are in progress in the lab.

Reviewer 1: How about clarifying that the triazole resistance of the C. haemulonii species complex used in this study is largely due to the increased expression of the efflux pumps and is less affected by the ERG11 mutation? If there is a report that the ERG11 mutation contributed significantly to the high resistance of the C. haemulonii species complex, please describe it in the discussion.

Authors: We agree with the reviewer’s comment. So far, no previous report has proved the significance of ERG11 gene mutations on the enzyme activity and\or the enzyme (the target) association with azoles. The authors only detected the presence of gene mutations. That is why we have pointed in the abstract session: ”Collectively these data pointed out the relevance of drug efflux pumps in mediating azole resistance in C. haemulonii complex, and mutations in ERG11p MAY contribute to this resistance profile.” Nevertheless, we have clarified the conclusion sentences in the revised version of the manuscript. To finalize, and just to reviewer knowledge, some of the ERG11 gene mutations detected in C. haemulonii are already detected in other Candida spp., including C. albincas, and for those yeast models a clear correlation between ERG11 gene mutation and resistance has been already traced.

Reviewer 2 Report

Well-designed study where the authors pointed out that the combination of
efflux pump activity and ERG11 missense mutations are associated in azole
resistance in clinical isolates of C. haemulonii. Comments- Did authors observe trailing effect while azoles MIC determination?

1.In table 2, authors should include MIC profile for C. tropicalis, C. krusei
and C. lusitaneae as efflux pump assay include data from these strains.

2.It would be nice if WT C. haemuloni can be included to compare
the efflux pump activity with clinical azole resistant C. haemuloni strains.

3. All genus/species name should be in italics throughout the manuscript.

Author Response

Reviewer 2: Well-designed study where the authors pointed out that the combination of efflux pump activity and ERG11 missense mutations are associated in azole resistance in clinical isolates of C. haemulonii.

Authors: The authors thank for the positive evaluation of the work.

Reviewer 2: Did authors observe trailing effect while azoles MIC determination?

Authors: Trailing effect is an important issue in the interpretation of the MIC results. However, our fungal isolates grow abundantly and well in high azole concentrations proving that the entire population is resistant. So, no trailing effect was observed.

Reviewer 2: In table 2, authors should include MIC profile for C. tropicalis, C. krusei and C. lusitaneae as efflux pump assay include data from these strains.

Authors: We are grateful for this valuable suggestion. We added an extra table on supplementary information as requested.

Reviewer 2: It would be nice if WT C. haemuloni can be included to compare the efflux pump activity with clinical azole resistant C. haemuloni strains.

Authors: C. haemulonii species complex are considered rare Candida species that until recently were erroneously identified, but they have emerged as (multi)drug-resistant isolates. Over the last few years, our group has collected about 30 isolates from different Brazilian hospitals in order to better characterize this fungal complex. So, it caught our attention that all these Brazilian isolates are both fluconazole and AMB-resistant strains. In this context, unhappily, we do not have any fluconazole susceptible strain in our sample collection, which precludes the execution of the experiment suggested by the reviewer. In fact, it was one of our original aims and to try to do it, we asked some researchers to kindly send us a fluconazole susceptible strain; however, I never obtained a positive answer. Just to the reviewer's knowledge, a simple glimpse into the available literature concerning the susceptibility profile of this fungal complex revealed that approximately 88% of C. haemulonii, 94% of C. haemulonii var. vulnera and 98% of C. duoboshaemulonii are resistant to fluconazole (we are preparing a review manuscript to explore this and other results in the resistance field). For all these reasons, we decided to include in efflux pump assay other non-albicans Candida species to make a reasonable comparison, but we agree that they are not (necessarily) the best one. Moreover, the susceptibility profile of all fungal species used in the present manuscript is summarized in supplementary tables.

Reviewer 2: All genus/species name should be in italics throughout the manuscript.

Authors: We are grateful for the correction. We modified the sentences in the revised version of the manuscript.